

# Total incombustible (mineral) content of *Cherax quadricarinatus* differs between feral populations in Central-Eastern Australia

Leyton J. Tierney[1,*], Clyde H. Wild[1] and James M. Furse[2,3,*]

[1] Environmental Futures Research Institute, Griffith University, Gold Coast, QLD, Australia
[2] Griffith Centre for Coastal Management, Griffith University, Gold Coast, QLD, Australia
[3] Miyazaki International College, Miyazaki, Japan
* These authors contributed equally to this work.

## ABSTRACT

*Cherax quadricarinatus* has been widely translocated within Australia, and a number of self-sustaining feral populations have established, and persisted, in central-eastern Australia for over 20 years: however, the biology and ecology of feral populations remain poorly understood. Using the loss-by-ignition method, this study investigated differences in the total content of incombustible material (as a proxy for total mineral content), between feral *C. quadricarinatus* populations in southeast Queensland and northeastern New South Wales. A total of 102 *C. quadricarinatus* were ignited, and percent total incombustible material was not proportional to the body size, or gender of the crayfish. Incombustible content was however, significantly different between some locations of capture (i.e., waterbodies). The site where incombustible content in crayfish was atypical, Lake Ainsworth, is a naturally acidic coastal lake, and we suggest that acidity and low concentration of calcium in that waterbody are likely responsible for the difference in mineral content detected in that population. Mechanism(s) driving the difference detected in the Lake Ainsworth population are unknown, but we suggest the acidic environment could directly impact maintenance of internal calcium reserves in the crayfish (intermoult), during recalcification of the cuticle (postmoult), or both. Limited calcium availability in the lake may also be a direct, or indirect, contributing factor. The ability of *C. quadricarinatus* to occupy acidic habitats while managing biomineralization challenges possibly could enable additional range-expansion of the species, and potential impacts on both endangered ecological communities and other biota occupying the acidic coastal habitats of Eastern Australia.

# INTRODUCTION

*Cherax quadricarinatus* (Von Martens) is one of 27 Australian species in the genus *Cherax* (additional species of *Cherax* are endemic to New Guinea) (*Furse et al., 2015*; *Lukhaup, Eprilurahman & Von-Rintelen, 2017*). The native range of *C. quadricarinatus* covers the

Corresponding author
James M. Furse,
j.furse@griffith.edu.au

tropical north of Queensland (Qld) and Northern Territory, and the southern parts of New Guinea (*Austin, Jones & Wingfield, 2010*; *Leland, Coughran & Furse, 2012*; *Riek, 1969*). Native range Extent of Occurrence (*IUCN, 2017*) in Australia is ~1.8 million km$^2$. The 2017 IUCN Red List status for *C. quadricarinatus* was Least Concern (*Austin, Jones & Wingfield, 2010*).

*Cherax quadricarinatus* is regarded as a hardy species, known to be tolerant of a broad range of environmental conditions (*Leland, Coughran & Furse, 2012*), for example, low oxygen levels, high temperatures, and salinity (*Saoud, Garza De Yta & Ghanawi, 2012*; *Wingfield, 2002*), and these are likely typical conditions over much of its native northern Australian distribution (*Jones, 1989*).

The species' native Australian range is characterized by distinct wet-dry seasons (the monsoon peak of the northern Australia wet season is December–March, *Bureau of Meteorology, Commonwealth of Australia (BOM), 2017c*), warm average temperatures year-round, with high air temperatures in summer: highest daily maxima >45 °C (*Bureau of Meteorology, Commonwealth of Australia (BOM), 2017a*). Northern Australia is also subject to periodic severe weather events including drought, heat waves and tropical cyclones. The native-range waterbodies occupied by *C. quadricarinatus* may be ephemeral, and inhabited by large predatory fish (e.g., *Lates calcarifer*; Bloch) and various *Crocodylus* species (including *Crocodylus* johnsoni Krefft, *Crocodylus novaeguineae* (Schmidt), and *Crocodylus porosus* Schneider). Overall the native range of *C. quadricarinatus* is a challenging environment for aquatic species.

Movements of *C. quadricarinatus*, well outside of its natural distribution, are due to its desirable biological characteristics, and aesthetics, making it well suited to aquaculture (*Lawrence & Jones, 2002*), research, the aquarium trade (*Furse et al., 2015*), and as a recreational fisheries target. *Cherax quadricarinatus* has been translocated both within Qld and to other regions of Australia (Western Australia and New South Wales (NSW)) for aquaculture, and illegally as a species for recreational fishing (*Doupé, 2007*; *Doupé et al., 2004*; *Furse et al., 2015*; *Leland, Coughran & Furse, 2012*). Additionally, the species has been translocated to more than 30 countries elsewhere, in regions including Africa, Asia, North America, Central America and Caribbean, South America, Europe, and Oceania (see list in *Furse et al., 2015*), and in a number of these cases the species has established feral populations (e.g., South Africa and Swaziland, Spain, Singapore, Mexico, Malaysia: *Belle et al., 2011*; *Gozlan, 2010*; *Naiqiddin et al., 2016*; *Nakata & Goshima, 2003*; *Nunes et al., 2017*; *Petersen et al., 2017*; *Vega-Villasante et al., 2015*).

While *C. quadricarinatus* is native to the tropics, successful aquaculture operations and feral populations have established, and persisted, in Australia, south of the Tropic of Capricorn (~23.40°S), and even further south into a montane area, with temperate climate, in southeast Qld. The southernmost extent of the species' present Australian distribution is Lake Ainsworth (NSW) at 28.78°S where a large feral population persists (*Leland, Coughran & Furse, 2012*).

*Cherax quadricarinatus* is the world's most intensively studied species of freshwater crayfish, but despite the body of knowledge on captive biology, physiology,
and aquaculture husbandry and production, the wild biology and ecology of the species remains poorly understood (*Furse et al., 2015*).

The species' ability to tolerate the at-times challenging environmental conditions within its native range evidently provides this species with the capacity, and considerable opportunity, to occupy the extensive and rather less environmentally challenging waterbodies elsewhere in Australia. Sufficient evidence is available to conclude that populations of this species are able to survive, and in some cases prosper, in atypical habitats and environmental conditions well outside its native range in Australia and elsewhere. Populations occupy municipal water reservoirs in areas of temperate climate, the subtropics, and tropics in both Hemispheres (e.g., *Ahyong & Yeo, 2007*), acidic coastal lakes in central-eastern Australia (*Leland, Coughran & Furse, 2012*), and waterbodies in the semiarid zone of Australia (J.M. Furse, 2014, unpublished data). However, it is not known if occupying these atypical habitats has any biological or physical impact(s) on *C. quadricarinatus*.

*Cherax quadricarinatus* is an ecologically-aggressive species (*Furse et al., 2015*; *Furse & Coughran, 2011*) and one consequence of escapees, or intentional releases, is that this species will establish and naturalize in many locations where it is likely to present a risk of disrupting/displacing the local fauna (*James et al., 2015*; *Leland, Coughran & Furse, 2012*; *Papavlasopoulou et al., 2013*). In particular this may threaten other species (including endangered species) (*Furse & Coughran, 2011*; *Leland, Coughran & Furse, 2012*; *Richman et al., 2015*), but can also cause damage to key or threatened habitats (*Horwitz, 1990*; *James et al., 2015*). Freshwater crayfish are keystone species (*Momot, 1995*) and capable ecosystem engineers (*Creed & Reed, 2004*; *Statzner et al., 2000*; *Statzner, Peltret & Tomanova, 2003*). Therefore, consequences of impacts caused by ongoing introductions, or spread, of *C. quadricarinatus* through waterways could be appreciable.

Specimens captured during a previous (2014) and unrelated distributional study of feral *C. quadricarinatus* populations in southeast Qld and northern NSW (J.M. Furse, *et al.*, 2014, unpublished data) presented an opportunity to investigate any differences between populations in distinctly atypical habitats.

The objective of this study was to determine if the total content of incombustible material in *C. quadricarinatus* differed between feral populations occupying different waterbodies within the study area.

## MATERIALS AND METHODS

### Collection and storage of crayfish

Crayfish were collected during a study over the Austral Summer–Autumn of 2014 (January–April). A total of 30 locations (i.e., waterbodies) were surveyed in southeast Qld and northeastern NSW, and crayfish were collected from five different waterbodies. The 30 waterbodies were selected for survey on the basis that self-sustaining feral populations of the species were known to have established there, or the species may have established there *via* translocation, migration through waterways, or overland movement. Crayfish were collected using a standardized trapping protocol (employing baited box traps, also called "bait" or "shrimp" traps, dimensions 250 × 250 × 370 mm, 75 mm

entrance apertures, ~2 mm mesh) that was applied at all locations. Trapping was primarily conducted in the morning (0630–1000) and afternoon (1300–1800), with a few late-evening and overnight sets deployed. Single measurements of water temperature, and pH were recorded at time-of-capture, at all locations (HI 98129, pH/Conductivity/TDS Tester; Hanna Instruments®, Woonsocket, RI, USA). Crayfish were collected under NSW Department of Primary Industries Fisheries Section 37 Permit P12/0026-2.0 and Qld Fisheries General Fisheries Permit 169932.

Long-term data were obtained from water-supply authorities, or literature, for the following parameters: water temperature, pH, total hardness (as $CaCO_3$, *NHMRC-NRMMC, 2011*), total alkalinity, and calcium and magnesium concentrations. These data covered capture dates of all crayfish in this study (except Lake Ainsworth), and in some cases covered the likely lifespan (i.e., 4–5 years) of those crayfish.

Following capture, crayfish were measured (Occipital-Carapace length) (OCL, *Morgan, 1997*), and gender (male, female, or indeterminate) was assessed by examination of external reproductive structures. Crayfish were placed in plastic Ziplock® bags, stored under refrigeration in the field, before being transported to the laboratory for euthanasia by freezing and subsequent frozen storage.

## Laboratory procedure

The percent total incombustible content of crayfish was determined using the Ash Content method of *Allen et al. (1974)*.

Crayfish were thawed at room temperature (~21 °C), and whole crayfish were dried to constant weight at 100 °C, preweighed to the nearest 0.0001 g (AND HR-300 analytical balance) and subsequently ground to a fine powder in a domestic coffee grinder (Sunbeam MultiGrinder II). Wherever possible the entire powdered crayfish were ignited (in preweighed crucibles), or in the case of the larger specimens: subsamples (average weight of subsample ~6.5 g). All powdered crayfish or subsamples were ignited for 1 h at 500 °C in an electric muffle furnace, cooled in a desiccator for 30 min before being weighed again. Following final weighing, all ignited samples were examined for any evidence of incomplete combustion (as per *Allen et al., 1974*). The ash residue of the ignited samples included all incombustible residues, that is, minerals, including metals, and other elements present in the crayfish, plus any uncombusted organic material bound within inorganic matrices, and hereafter is collectively considered as a proxy measure of the mineral content of the samples. The percent dry weight total incombustible material (hereafter incombustible content) of each crayfish was determined by calculation.

## Data analysis

Raw data were plotted (scatter and boxplots) and examined for obvious trends. Effects of gender and location were assessed simultaneously by factorial ANOVA, and final analysis used a main effects ANOVA with gender, OCL and location as factors. A *post hoc* Tukey's LSD provided *p*-values for differences between locations. All analysis were performed using STATISTICA for Windows (Version 7.1) (StatSoft, Tulsa, OK, USA),
**Table 1 Summary of capture locations, environmental conditions at time of capture, and crayfish captured in this study.**

| N = 102 | | Location of capture | | | | |
| --- | --- | --- | --- | --- | --- | --- |
| | | Lake Ainsworth (NSW) | Lake Dyer (Qld) | Emigrant Creek Dam (NSW) | Lake Moogerah (Qld) | Lake Somerset (Qld) |
| Coordinates | | 28.783500°S 153.591852°E | 27.628950°S 152.374936°E | 28.769090°S 153.517838°E | 28.041717°S 152.551186°E | 27.067288°S 152.588607°E |
| Water temperature (°C) (at time of capture) | | 28.5 | 25.4 | 24.4 | 31.7 | 27.1 |
| pH at location of capture (time of measurement) | | 6.43 (1420) | 8.80 (1630) | 7.13 (0905) | 8.91 (1730) | 8.08 (1000) |
| Total number of crayfish captured | | 72 | 4 | 3 | 2 | 21 |
| Gender | Male:Female | 30:42 | 1:3 | 3:0 | 0:2 | 9:11 |
| | Indeterminate | – | – | – | – | 1 |
| OCL (in mm) | Maximum | 64.22 | 50.13 | 59.79 | 60.51 | 67.41 |
| | Average | 47.00 | 45.71 | 40.17 | 57.67 | 46.76 |
| | Median | 46.37 | 45.81 | 52.51 | 57.67 | 44.98 |
| | Minimum | 20.64 | 41.09 | 8.20 | 54.82 | 26.48 |
| Percent DWT incombustible material mean and ±95% C.I. values (in g) | Upper 95% C.I. | 42.93 | 57.80 | 53.13 | 132.82 | 51.16 |
| | Mean | 41.79 | 46.11 | 50.80 | 48.58 | 48.96 |
| | Lower 95% C.I. | 40.66 | 31.42 | 48.47 | −35.66 | 46.77 |

Note:
Refer Fig. 2 for boxplot associated with percent DWT incombustible content mean and ±95% confidence interval (C.I.) values reported here.

critical value for α was 0.05 throughout. Long-term water temperature and water chemistry data were tabulated to show any differences between locations.

## RESULTS

Crayfish were collected at all times of the day, from waterbodies within the subtropical climate zone and a temperate-classified area of central-eastern Australia (climate zones based on a modified Koppen system, *Bureau of Meteorology, Commonwealth of Australia (BOM), 2017b*), with four of the collection locations being large municipal water reservoirs on basaltic strata, and one (Lake Ainsworth) a perched coastal-dune lake underlain by siliceous sand deposits (*Timms, 1982*). Well-established biotic communities were evident at all locations, and included: algae, macrophytes, invertebrates, and vertebrates (fishes, reptiles, birds): typically, crustacean zooplankter's were highly abundant.

A total of 102 crayfish were collected from the five locations (Table 1), processed and ignited: 58 females, 43 males and one case where gender could not be assigned. The case where gender could not be assigned was excluded from statistical analysis including gender as a factor. All crayfish were captured in fully-calcified (i.e., intermoult) condition, both clean and very dirty exoskeletons were noted, indicating recent moult activity, or otherwise.

**Table 2 Long-term water temperature and water chemistry data at locations of capture: average values highlighted in bold.**

| | | Location of capture | | | | | |
| --- | --- | --- | --- | --- | --- | --- | --- |
| | | Lake Ainsworth[A] (NSW) | Lake Ainsworth[B] (NSW) | Lake Dyer (Qld) | Emigrant Creek Dam (NSW) | Lake Moogerah (Qld) | Lake Somerset (Qld) |
| Water temperature (°C) | Maximum | – | 30.90 | – | 28.00 | 22.97 | 28.90 |
| | Average | – | **23.70** | – | **20.90** | **22.50** | **21.56** |
| | Median | – | 24.70 | – | 21.50 | 22.40 | 22.50 |
| | Minimum | – | 14.90 | – | 13.30 | 22.04 | 13.70 |
| pH | Maximum | 9.40 | 8.00 | – | 9.60 | 7.75 | 8.40 |
| | Average | **6.20** | **5.91** | – | **6.89** | **7.71** | **7.56** |
| | Median | 6.20 | 5.81 | – | 6.90 | 7.72 | 7.50 |
| | Minimum | 3.50 | 4.00 | – | 6.10 | 7.64 | 6.90 |
| Total hardness as $CaCO_3$ (mg $L^{-1}$) | Maximum | 219.00 | 22.90 | – | 37.00 | 109.00 | 64.00 |
| | Average | **35.24** | **22.76** | – | **16.70** | **86.60** | **50.31** |
| | Median | 22.40 | 22.50 | – | 16.00 | 83.00 | 50.00 |
| | Minimum | 5.99 | 22.10 | – | 6.00 | 64.00 | 28.00 |
| Total alkalinity as $CaCO_3$ (mg $L^{-1}$) | Maximum | – | – | – | 39.00 | 101.00 | 78.00 |
| | Average | – | – | – | **23.87** | **81.10** | **51.32** |
| | Median | – | – | – | **23.00** | 78.50 | 52.00 |
| | Minimum | – | – | – | 12.00 | 66.00 | 21.00 |
| Calcium (mg $L^{-1}$) | Maximum | 62.40 | 2.60 | – | 8.40 | – | 13.00 |
| | Average | **14.2** | **2.21** | – | **2.97** | – | **9.23** |
| | Median | 1.4 | 2.40 | – | 2.90 | – | 9.00 |
| | Minimum | 0.6 | 0.80 | – | 1.00 | – | 5.50 |
| Magnesium (mg $L^{-1}$) | Maximum | 15.12 | 4.04 | – | 14.00 | – | 8.00 |
| | Average | **3.96** | **4.20** | – | **2.30** | – | **6.12** |
| | Median | 2.40 | 4.15 | – | 2.20 | – | 6.05 |
| | Minimum | 1.08 | 4.87 | – | 0.90 | – | 3.50 |

**Notes:**

Data from Emigrant Creek Dam and Lake Somerset encompass the likely lifespan of the crayfish in this study. No data available for Lake Dyer. Unless otherwise stated, data provided by water-supply authorities.

Lake Ainsworth[A], data from *Timms (1982)*. Lake Ainsworth[B], data from *Kadluczka, Howells & Van Senden (1996)*.

Emigrant Creek Dam, weekly samples: January 2009–May 2017.

Lake Moogerah, single values for Temperature, pH (March 2012), Hardness (November 2011), irregular monthly samples for Alkalinity (2011–2013).

Lake Somerset, weekly or monthly samples: February 2009–May 2014 (Temperature: July 2011–May 2014).

Mean & median values for $CaCO_3$ and calcium in Ainsworth[A] data indicate high-value outlier-induced skew, we therefore draw on Ainsworth[B] data.

Water temperature and pH at time of capture differed between locations of capture (Table 1). On average, long-term water temperatures were similar between locations, but some differences in water chemistries were evident between locations (principally pH, total hardness, calcium, and magnesium (Table 2). Water chemistries at Lake Ainsworth and Emigrant Creek Dam were comparable, but Emigrant Creek Dam had, on average, the lowest values in this study for hardness (16.70 mg $L^{-1}$) and magnesium (2.30 mg $L^{-1}$). On average, Lake Ainsworth had the lowest values for pH (5.91) and

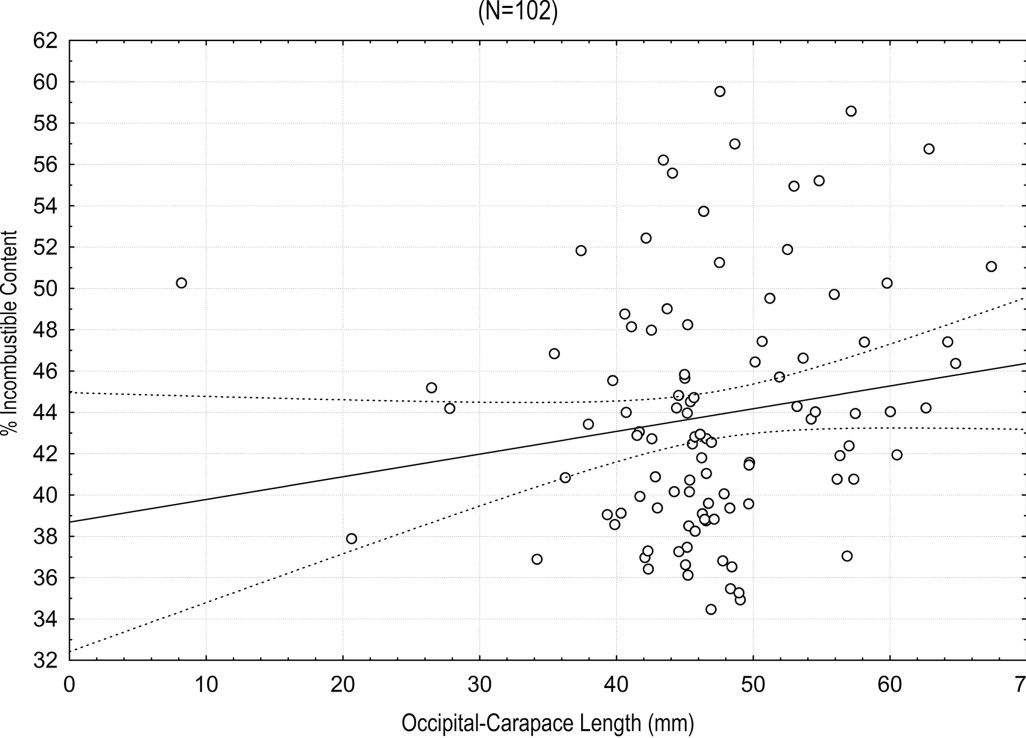

Percent Incombustible (Mineral) Content *versus* Occipital-Carapace Length in *Cherax quadricarinatus* (N=102)

**Figure 1 Percent DWT incombustible content of *Cherax quadricarinatus* vs size for all 102 specimens ignited in this study.** The incombustible content of crayfish was not proportional to body size.                                     

calcium (2.21 mg L$^{-1}$), and hardness (22.76 mg L$^{-1}$) was substantially lower than both Lakes Moogerah (86.60 mg L$^{-1}$) and Somerset (50.31 mg L$^{-1}$).

A histogram of the raw incombustible content indicated a slight positive skew. A log(x) transformation was applied and rendered the distribution of the data approximately normal. Percent incombustible content of crayfish plotted *vs* OCL (Fig. 1) showed a weak, nonsignificant positive correlation ($R^2 = 0.027$, $p = 0.10$), indicating that correction for OCL was not required for subsequent analysis. There was no significant difference in incombustible content between gender ($p = 0.39$). The gender*location interaction term was also not significant ($p = 0.97$), but location of capture was significant ($p = 8.0 \times 10^{-8}$), all by two-way factorial ANOVA.

A boxplot of percent incombustible content by location of capture indicated which locations differed (Fig. 2). The wider 95% confidence intervals (C.I.) for the Lake Moogerah data can be attributed to the natural variation (13% difference) of incombustible content in the smaller ($N = 2$) sample size at that location, however, these data did not unduly influence, or leverage, any subsequent statistical analysis. A LSD *post hoc* test indicated Lake Ainsworth as significantly different to Emigrant Creek Dam and Lake Somerset ($p = 0.0031$ and $p = 5.0 \times 10^{-7}$, respectively). Incombustible content did not differ significantly between any other locations.

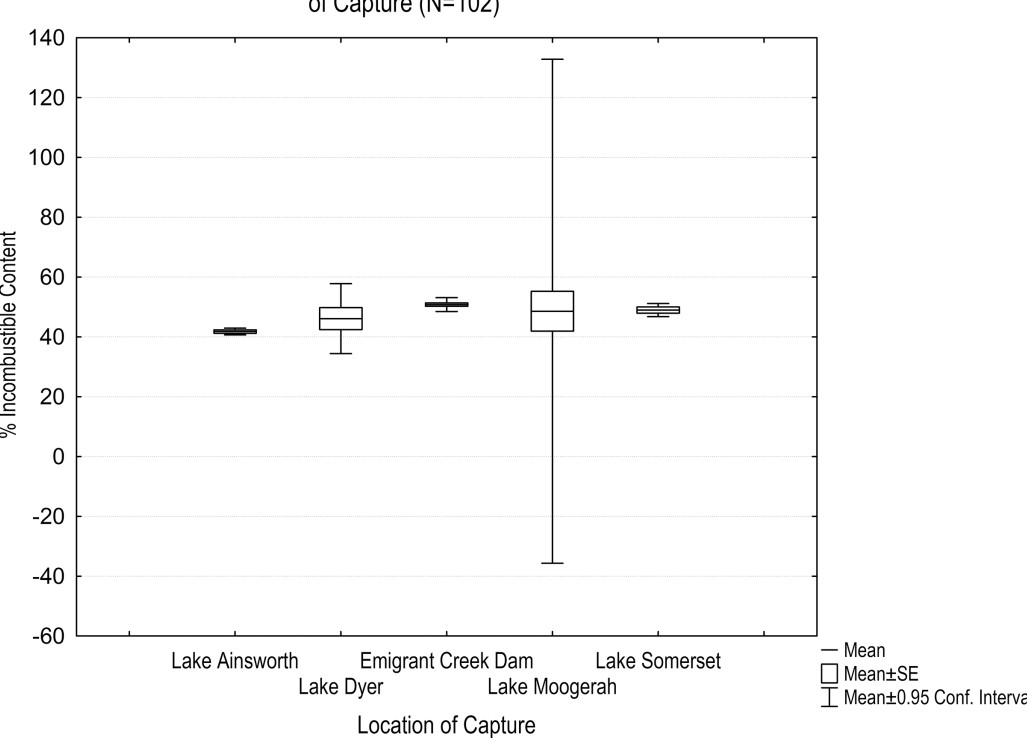

Percent Incombustible (Mineral) Content of *Cherax quadricarinatus versus* Location of Capture (N=102)

**Figure 2 Boxplot of percent DWT incombustible content of *Cherax quadricarinatus vs* location of capture.** Specimens from Lake Ainsworth had significantly lower total incombustible content compared to Emigrant Creek Dam and Lake Somerset. Refer Table 1 for mean, and ±95% confidence interval (C.I.) values associated with this boxplot.

## DISCUSSION

This study advances our understanding of wild populations of *C. quadricarinatus* within Australia, and information uncovered in this study is relevant to both the ongoing translocations (and spread) of this species within Australia, but also in other regions elsewhere in the World.

It is well established that *C. quadricarinatus* is a "tough" species of freshwater crayfish (see review in *Furse et al., 2015*), tolerant of environmental extremes. However, the present study has uncovered a new dimension of the capacity of *C. quadricarinatus* to cope with challenging environmental conditions. In this case crayfish were collected from established, self-sustaining populations that have persisted well outside this species' native-range for more than 6 years (e.g., Lake Ainsworth) and over 20 years in the case of Lake Somerset. Similarly, all waterbodies supporting these populations were distinctly non-native habitat-types located in non-native climatic zones (e.g., temperate to subtropical climates) (Modified Koppen classification, *Bureau of Meteorology, Commonwealth of Australia (BOM), 2017b*).

Water chemistry data indicated the waterbodies "grouped" in two categories: locations in NSW with lower pH and hardness, and locations in Qld with neutral pH and considerably higher hardness. Similarities in water chemistries between Lake

Ainsworth and Emigrant Creek Dam may, in part, be attributable to that region's somewhat uniform basaltic geology, and both locations being situated in the same catchment, ~7 km apart.

While it has been previously reported that *C. quadricarinatus* can occupy acidic environments (*Leland, Coughran & Furse, 2012*, Lake Ainsworth: a naturally acidic coastal window lake, *Timms, 1982*), any long-term physical/physiological impact(s) of this species occupying such habitat are not well documented. This study establishes that there is a demonstrated physical impact on *C. quadricarinatus* occupying acidic habitat, in this case a significantly lower total incombustible-mineral content compared to crayfish collected from other nonacidic habitats. In this case, we presume the incombustible content was primarily composed of calcareous minerals derived from the crayfish exoskeletons and tissues, and any other residues (e.g., other metals and incombustible nonmetals) were at negligible levels (*sensu Jussila, Henttonen & Huner, 1995*).

Despite acidic conditions, and in particular having to maintain a calcified exoskeleton (i.e., internal stores of calcium) in adverse conditions, this study supports a conclusion that populations of *C. quadricarinatus* can persist, and evidently prosper, medium to long-term in acidic habitats, including where calcium is not abundant. An alternative interpretation is that this study has uncovered evidence indicating the *C. quadricarinatus* at Lake Ainsworth are not actually coping with the pH and low mineral availability, and the population is under-mineralized. This alternative explanation seems less likely given that *C. quadricarinatus* has prospered and remained so abundant in Lake Ainsworth for nearly a decade.

Other Australian *Cherax* are known to occupy acidic habitats; for example, *Cherax cuspidatus* Riek (Lake Ainsworth, native range-and habitat) (see *Leland, Coughran & Furse, 2012*), and *Cherax cainii* Austin and Ryan (formerly *Cherax tenuimanus* (Smith)) (old mining pits, native range, non-native habitat) where mild stress due to acid exposure was reported (*Storer, Whisson & Evans, 2002*).

The exact mechanism driving the lower incombustible content in the Lake Ainsworth population (or conversely the higher incombustible content in other waterbodies) in this study is unclear, but given the acidic condition of Lake Ainsworth, we speculate it is possible that acidity is a factor directly, or indirectly, contributing to the differences reported here.

Biomineralization is a reasonably well-understood phenomenon in the Crustacea and freshwater crayfish, so some speculation that effect(s) of an acidic environment may occur in *C. quadricarinatus* is appropriate. However, we hypothesize that other aspects of the water chemistry of Lake Ainsworth may also be impacting mineralization in the *C. quadricarinatus* population.

The mineral that is principally responsible for providing rigidity and strength in the structure of the exoskeleton of freshwater crayfish is calcium carbonate ($CaCO_3$) (*Huner, Kowalczuk & Avault, 1978*; *Reynolds, 2002*). Calcium is therefore one of the key elements for growth in freshwater crayfish (*Reynolds, 2002*): especially so postmoult (*Greenaway, 1985*; *Huner, Kowalczuk & Avault, 1978*). Postmoult, part of the mineral requirement for recalcifying the new cuticle is met by retaining calcium from the previous

exoskeleton (*via* storage in the gastroliths located in the foregut, formed premoult) with the remainder derived from the water column, food, or other ingested material such as the exuvium (*Adegboye, Hagadorn & Hirsch, 1975*; *Greenaway, 1985*).

Calcareous minerals are a key structural component of crustacean exoskeletons (*Bentov et al., 2012*), and the acidity of Lake Ainsworth (the most acidic in the study) has the potential to directly (and indirectly) influence calcareous mineral content in the population of *C. quadricarinatus*. The acidity of the Lake may directly influence, *via* increased solubility, internal calcareous mineral content premoult (i.e., during decalcification and gastrolith formation), but also postmoult, during both recalcification from internal stores, that is, solubilization reabsorption of gastrolith calcium) and uptake from external sources of calcium. Availability of bicarbonate ($HCO_3^-$) is reduced in acidic conditions and if insufficient bicarbonate is available from internal sources, this may also directly contribute to reduced internal calcareous mineral content.

It is therefore possible the pH of Lake Ainsworth is, at times, simply outside of the biologically-tolerable range where *C. quadricarinatus* is able to calcify, or maintain calcification.

Water chemistry data indicates Lake Ainsworth is not a calcium-rich environment, and limited calcium availability may directly influence the mineral content of *C. quadricarinatus* in the lake. Lower calcium availability may also interact with acidity of the lake, possibly indirectly creating a limiting condition (*Reynolds, 2002*), or becoming a limiting factor. One possible scenario would be the acidity of Lake Ainsworth (i.e., reduced carbonate availability), interacting with the low calcium levels thus limiting availability of both the components of calcium carbonate. While water chemistry provides potential explanations for the difference in total mineral content, assessed *via* incombustible content, at Lake Ainsworth, it is possible other unknown (and unmeasured) biotic and/or abiotic factors may be responsible.

It is not possible to determine why incombustible content in the crayfish from Emigrant Creek Dam was significantly higher than Lake Ainsworth despite lower hardness at that location. We speculate that the higher pH and calcium values, and lower magnesium value, at Emigrant Creek Dam may potentially negate any effect of the lower hardness.

Irrespective of the mechanism, the results reported here are consistent with those of others workers reviewing (*France, 1983*, *1987*; *Greenaway, 1985*) or investigating physical, or growth-or weight-related impacts of acidic environments on freshwater crayfish (e.g., *France, 1983* pH 5.35; *France, 1987* pH 5.4–5.6; *Haddaway et al., 2013* pH 6.5, 7.1, and 8.6; *Malley, 1980* pH 4.0–6.7; *Yue et al., 2009* pH 6.8).

Increasing atmospheric $CO_2$ may increase acidity in the acidic habitats of Eastern Australia, potentially impact rare and endangered native crayfish (*Cherax* and *Tenuibranchiurus* spp.) (*Beaune et al., 2018*), but also feral populations of *C. quadricarinatus* in those habitats. Ongoing surveys and monitoring of these feral *C. quadricarinatus* populations may track any changes in the distribution of *C. quadricarinatus* but may also permit the mechanism(s) driving the differences in incombustible-mineral content reported

here to be investigated and documented, in particular any effects of Lake Ainsworth water chemistry on the mineralization and calcareous content of *C. quadricarinatus.*

## Potential management implications

We speculate that the ability of *C. quadricarinatus* populations to establish and persist, and maintain internal calcium levels, in calcium-poor and acidic conditions, could have important management implications. For instance, the substantial resident population of *C. quadricarinatus* in the calcium-poor Lake Ainsworth, needs to somehow obtain the calcium (*sensu Adegboye, Hagadorn & Hirsch, 1975*) necessary for maintaining adequate internal levels. Other biota occupying the lake, especially calcified biota (i.e., other native crayfish), are likely be an accessible and useful source of calcium for *C. quadricarinatus* occupying the Lake.

*Leland, Coughran & Furse (2012)* reported the smaller and ubiquitous *C. cuspidatus* appeared to be absent from the lake, but not from the adjacent, acidic, *Melaleuca quinquenervia* swamps and associated roadside drains. Monitoring of the adjacent habitats by the authors confirms that situation remains unchanged and while *C. cuspidatus* is present, *C. quadricarinatus* has not been detected in these nearby acidic habitats. It is unclear if *C. quadricarinatus* is involved in the apparent absence of *C. cuspidatus* from the lake, but it is reasonable to suggest the smaller native crayfish (and other calcified biota) have been a relatively available and useful source of calcareous minerals for the larger *C. quadricarinatus, via* predation and consumption.

Eastern Australia's extensive acidic coastal *Melaleuca* swamps extend, discontinuously, from the north of Qld to Sydney in the south, and include a number of coastal window lakes similar to Lake Ainsworth. These acidic coastal swamps and lakes are occupied by other rare and/or critically endangered species (including other species of crayfish: e.g., *Tenuibranchiurus* spp. *Coughran, Dawkins & Furse, 2010*; *Dawkins et al., 2010, 2017*), and include endangered ecological communities (e.g., freshwater wetlands on coastal floodplains in NSW). It is possible the ability of *C. quadricarinatus* to occupy and survive in these acidic habitats could potentially lead to direct, or indirect, negative impacts on the various rare and endangered native floral and faunal species in these habitats.

Similarly, while not continuous, the coastal swamps of eastern Australia could potentially act as an "invasion" route that *C. quadricarinatus* could exploit, possibly providing an otherwise unavailable pathway ~600 km south, or >1,500 km to the north and west (adjacent the species native range).

## Limitations of study

Loss-by-ignition residues may include nontargets (e.g., metals and other incombustible elements) and possible small traces of uncombusted organic material, and this presumably applies to the incombustible material data obtained in this study. Analysis of mineral content, or other constituents of the combustion residues, was not possible in this study, however, it is reasonable to assume residues were principally mineral, and any other incombustible nonminerals were minor components

(*Bryan, 1967*; *Jussila, Henttonen & Huner, 1995*). Therefore, our standard method, applied to all samples, has resulted in a consistent measure of actual minerals, plus all nontargets, in all samples. Although numbers of crayfish collected from some locations were small ($N < 20$), most 95% C.I. were tight and the effects of location of capture on incombustible content were strong. As previously outlined, the broader 95% C.I. at Lakes Moogerah and Dyer are attributable to natural variation in the incombustible content in the crayfish captured there (13% and 1.6%, respectively): results need to be interpreted accordingly until the mineral content in these populations has been investigated and clarified.

Recent water chemistry data, for the parameters of interest in this study, do not exist for Lake Ainsworth; the historical water chemistry data available to us did however, permit satisfactory characterization of the lakes water chemistry, in particular: establishing low $CaCO_3$ and calcium concentrations, and persistent acidic conditions. The 1987 Lake Ainsworth maximum, average and median values for $CaCO_3$ and calcium indicated high-value induced skew, but as values for $CaCO_3$ and calcium in the 1996 data were unremarkable, we drew on the more recent 1996 data.

## CONCLUSIONS

The incombustible content of feral *C. quadricarinatus* populations differed significantly between the locations in this study, and the low total mineral content (using incombustible material as a proxy for mineral content) in crayfish from Lake Ainsworth is likely attributable to the lake's pH and limited calcium availability. This study supports other findings that *C. quadricarinatus* is tolerant of environments that are acidic, but also calcium-poor, and additionally demonstrates that despite these conditions the species can evidently prosper, albeit with some physical impact. This opens the possibility of the species having the potential to occupy many or all of the extensive coastal swamps of central-eastern Australia, with possible implications for the resident endemic flora and fauna.

## ACKNOWLEDGEMENTS

The study was conducted as 3rd year Undergraduate Project by Leyton J. Tierney under the supervision of James M. Furse and Clyde H. Wild. Many thanks to Patrick Plunkett (NSW DPI Fisheries Officer) for accommodating our numerous and ongoing trapping activities in Lake Ainsworth and environs. Shane Norman Patrick Archibald-Howard kindly facilitated access to some water chemistry data. Thanks are extended to the following for kindly providing historical water chemistry data: Stuart Hood (Rous County Council, NSW, Emigrant Creek Dam), Ballina Shire Council (NSW, Lake Ainsworth), and Seqwater (Qld, Lakes Moogerah and Somerset). The authors appreciate the suggestions and comments from our Editor and three anonymous reviewers.

### Funding

The study was supported by The Griffith School of Environment (Gold Coast campus), the Griffith Centre for Coastal Management and the Environmental Futures Research Institute

(Griffith University). There was no additional external funding received for this study. The funders had no role in study design, data collection and analysis, decision to publish, or preparation of the manuscript.

## Competing Interests

The authors declare that they have no competing interests.

## Author Contributions

- Leyton J. Tierney conceived and designed the experiments, performed the experiments, analyzed the data, prepared figures and/or tables, authored or reviewed drafts of the paper, approved the final draft.
- Clyde H. Wild conceived and designed the experiments, analyzed the data, contributed reagents/materials/analysis tools, prepared figures and/or tables, authored or reviewed drafts of the paper, approved the final draft.
- James M. Furse conceived and designed the experiments, analyzed the data, contributed reagents/materials/analysis tools, prepared figures and/or tables, authored or reviewed drafts of the paper, approved the final draft, collected the crayfish.

## Field Study Permissions

The following information was supplied relating to field study approvals (i.e., approving body and any reference numbers):

Crayfish were collected under NSW Department of Primary Industries Fisheries Section 37 Permit P12/0026-2.0 and Qld Fisheries General Fisheries Permit 169932.

## Data Availability

The raw data are available in a Supplemental File.

## Supplemental Information

Supplemental information for this article can be found online at http://dx.doi.org/10.7717/peerj.6351#supplemental-information.

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
