# Peer review of "Total incombustible (mineral) content of Cherax quadricarinatus differs between feral populations in Central-Eastern Australia"

_PeerJ, doi:10.7717/peerj.6351_

## Round 0.1 · original submission · Major Revisions

· Academic Editor

Major Revisions

Thank you for submitting your work to PeerJ. You will find three thorough reviews of your manuscript, with two reviewers recommending Major Revisions and a third recommending Rejection. I have decided on the recommendation of Major Revision if the serious concerns from reviewer 3 regarding the experimental methods used can be addressed.

All three reviewers reached general agreement on several points that should be addressed in any resubmission. These general concerns were:

- The manuscript strays from the central question being tested. The manuscript should be shortened and reorganized to clearly state the research question and how it is addressed by the data. In particular some reviewers noted that the data do not support broad statements about the spread of invasive crayfish. Extraneous information should be removed and limitations of the approach used and data collected should be addressed in a resubmission.

- A lack of information from the collection sites. pH was measured at collection sites, but reviewers noted the need for several other variables, and expressed concerns about weather station data being used as a proxy for water body temperature. Reviewers agree that the data in Table 1 are not relevant to the study.

- More information about pH measurement needs to be added to Table 2, along with other environmental variables that could affect data interpretation (as detailed by the reviewers).

- Figure 1 as currently presented is not necessary (reviewers 1 and 2), or should be expanded to include more information about feral populations (reviewer 3). Reviewers had questions about data presentation in figure 2 and 3. Two reviewers thought figure 3 should be removed. The small sample size from Lake Moogerah and its effect on figure 4 was noted, as was the presentation of statistical significance in this figure.

- Two reviewers note the need for added background references and information.

All reviewers felt that with reorganization and additional data on collection sites the work presented would meet PeerJ editorial guidelines for basic reporting and validity as manuscripts are not judged on potential impact. However, the most serious criticism that led to the recommendation to Reject by reviewer 3 surrounds the loss-by-ignition method used to measure mineral content in collected specimens. The concerns of this reviewer would need to be addressed to meet PeerJ’s requirement that all experiments be conducted to a high technical standard. The recommendation from all reviewers that additional environmental information would be needed to interpret data on calcium content might also affect the validity of the findings.

I invite you to consider the valuable feedback from all three reviewers and resubmit a revised manuscript along with a letter explaining how this feedback was addressed. I should note that several reviewer comments suggest that additional experiments and/or field data would be needed. You would need to clearly explain how these concerns have been addressed in a resubmission.

Reviewer 1 ·

Basic reporting

The manuscripts is based on a study of limited importance, though it has its merits as part of the freshwater crayfish research. My suggestion is a major revision, and compacting the manuscript in the format of a short note.

Experimental design

Nothing fundamentally wrong with the experimental design. I would have wanted to read a rationale for why the authors choose these sampling sites.

Validity of the findings

I found the data insuffienctly presented, with at least these aspects to be improved 1) the background data from the sampling sites should include sufficient water quality details, including Ca and other cations relevant for calcium uptake; 2) details regarding population density, growth rates, etc., which could explain some of the differences been discovered; 3) Lake Moogerah sample should be omitted (N=2!) or statistically treated to fit the overall sampling intensity, with the same done for other sites, too, if found relevant. This is due to the sample size differences. This should be evident based on the Fig. 4; 4) Fig. 3 does not seem relevant, as it only indicates that this set of data does not explain any relationship in the data set. This could be treated in body of the text; 5) is Fig. 2 combining the whole data set, i.e., samples from different locations? If yes, then this pooling of the data should be better discussed. If not, then what is in Fig. 2?; 6) Table 1 presents climatic details, been obtained from different periods of time. Do the differing data collection periods affect the variation seen in the data? Could there be time period related artifact created here?

Additional comments

I would recommed compacting the manuscript and making into a short note. Some suggestions regarding data presentation improving presented in the boxes above. After working with data volume and presentation, the authors should rework with the Discussions section, as there should now be more food for speculations.

Reviewer 2 ·

Basic reporting

Majority of the article is well written, using good grammar with professional English language. However, the title of the article is broad and verbose. It can be easily modified and can be focused on the research question.
Introduction is well written but is out of focus. It reads like a good literature review on the basic ecology of the freshwater crayfish but does not say enough on the main aim of this article which is difference in ash (mineral) content of two crayfish populations.
Some of the terms like ‘physiochemical characteristics of waterbodies’ and ‘physiology of crayfish’ are very broad terms and are not part of this research. The mineral content of the species and recording of pH and temperature do not qualify for physiology and physiochemical characters.
The introduction section needs to be reduced to forty percent of its length by putting greater focus on the research question and justifying this research.

Experimental design

Experimental design of this research is simple and does not need much elaboration. However, more information on the number and time of crayfish captured from each location will be helpful. What was the moult stage of crayfish captured? The mineral content of the crayfish is dependent on the moult stages.

Validity of the findings

The findings of this research are good and valid. Our knowledge of chemistry tells us that pH has a profound impact on minerals, mainly on their biological availability.
We also know crustaceans, including freshwater crayfish are highly adaptive and over a period of time they can thrive very well in harsh environmental conditions.
Freshwater crayfish in Western Australia are also known to live in acidic waters filled in discarded mining voids.

Additional comments

Table 1 is irrelevant as it describes the location and temperature data recorded from the weather station.
The impact of these air temperatures on the temperatures of the sampled water bodies is not clear. Though coordinates are provided, it is hard to estimate the distance from the weather station to the sampled waterbodies. Moreover, as temperature is influenced by the volume and depth of waterbodies, the presence of table 1 does not add, in anyway, the science to this article. Therefore, please delete table 1.
Please modify table 2. pH is dependent on the sampling time of the day and natural productivity of the waterbody. Please provide the time when pH was recorded and any information on the presence of life (micro algae, plants, zooplanktons etc.) in these waterbodies.
It seems that a total 162 animals were trapped. Why sample size from every waterbody is not provided. It would be easy to understand if data are presented in the format of ‘mean with SE’ and with n from each waterbody.
Figure 1 – delete. The reference to native range and distribution of red claw should be sufficient in the text.
Figure 2 is relevant. However, legend needs more details. N?
Figure 3 – delete. As there was no relationship between length and mineral content, there is no need to include this figure.
Figure 4 is relevant and is directly related to the theme of this article.

Discuss also hovers along with those discussion points which are not related to the aim of this research. The aim of this research should focus on the variation in the mineral contents of the different populations of the freshwater crayfish and its relationship with the pH of the waterbodies (we don’t know the pH of all water bodies). Only then, it can include some speculations about the relevance and justification of the research to impacts of translocation and not in reverse direction.
Conclusion is well stated but it does incorporate some information which is of common knowledge and does not come from this research.
Please spend some more time in revising this article and try to be focused on the research question. Also include some more references from Western Australia, where it is reported that the local crayfish population has made home to various water bodies very different from where they originally come from.

Reviewer 3 ·

Basic reporting

The writing in this manuscript is on the whole clear and succinct. I especially appreciate the engaging tone and varied sentence structure, which easily guides the reader along.

Here are some comments, provided in the structure suggested by PeerJ:


Professional English:

Please provide appropriate punctuation for a few instances of run-on sentences (eg lines 33, 126, 182, 190, 195, 204).


Professional article structure:

The organization of some sections would benefit from some reworking or addition of information Of most importance to rectify are the introduction and discussion. The introduction does not include a broad background section in which to place this piece of research. Based on the data presented, I would have appreciated an introduction to the topic of nonnative species and means by which they persist in conditions unlike that of their native range, including specific examples. There is also extraneous information presented in the introduction that I did not find useful, such as climate data in parts of Australia. The discussion could be more developed to both create a better flow of ideas; as is, there are some jumps in topic between paragraphs.

L51: Please begin with a more general introduction for the research that is to be presented. This introduction, for example, might touch upon ways other organisms have found success in nonnative habitats, despite substantial differences in abiotic or biotic factors from their home ranges. This seems like the main theme of your paper. It might also be worthwhile to discuss what benefits crayfish or other crustaceans receive from producing mineralized exoskeletons (e.g. predator defense) so that you may explain the functional importance for your findings later.
L60-64: As this paragraph stands, the specific link to the research is unclear. Do these weather conditions control the water parameters of the lakes?

L134-135: Again, it would be useful if there was more buildup to this objective, especially in relation to why one might expect there to be differences in mineralogy across these different water bodies and why mineralogy was chosen as the defining feature of these different populations to study.

L178-179: The sample size is more appropriate for the methods.

L162-167: Please incorporate this in the introduction, as the additional information about the lake substratum forms part of background information

Please consider building larger paragraphs in the results (lines 182- lines 200) and discussion (lines 254-270) to better construct a flow of ideas.



Literature references and background:
There has been a great body of literature built up around the structure and composition of crustacean exoskeletons over the last few decades. The literature cited in your discussion would be bolstered by the addition of some of these works. As such, some of the main information cited is backed by a textbook, not a primary piece of literature. I would recommend reading a few of the classics for the best overview, such as the Raabe papers I cited; Bentov has some well-written articles on some specialized regions of the crayfish exoskeleton too.

L130: Is this study related to this research? If so, please cite it fully. If not, consider removing this extraneous information.

L236-238: The exoskeleton of crayfish, like most crustaceans, is not primarily mineral; instead, it is based of chitin and protein fibers impregnated with calcium carbonate and calcium phosphate (Bentov et al., 2012; Raabe et al., 2006; Raabe et al., 2005). As such, calcium is likely not the most important element; an exoskeleton can still harden to some degree after molting through the tanning process. Consider rephrasing to state that it is an integral exoskeleton element without overstating its importance.

L247-257: Please make the connection between exoskeleton structure and composition to provide the relevance for including this study.

L246: This is a great spot to discuss one of the major impediments to calcification at lower pHs that has not yet been addressed in this manuscript: the reduced availability to carbonate ions. As pH decreases, the carbonate system shifts from a predominance of carbonate to bicarbonate and then carbonic acid. Crustaceans are believed to have some mechanisms to procure carbonate ions at the site of calcification, but the small amount available externally may be playing a role in the lower amounts of calcification at lower pHs observed here.



Figures and tables:
Table 1: Consider cutting. These data have not established relevance to this study.

Table 2: Please note what scale pH is measured in. It would also be useful to know what pH ranges this species experiences in its native habitat, either provided here in this table or in the text.

Figure 1. As this research focuses on the locations where these crayfish are established outside of their range, this figure could be improved by also showing the study locations to give readers not only a sense of the feral extent of these crayfish but highlight the spatial range of locations studied.

Figure 2. It is a bit unusual to see a histogram in manuscript, and this might not be necessary because readers are able to get a sense of the number of crayfish with a particular mineral composition in both figures 3 and 4.

Figure 3 and 4: Specify what mineral content is measured here, as other minerals might be magnesian calcite, for example, which was not measured. Also, please state the major finding or trend of the analysis for both Figure 3 and 4 in the caption. For Figure 4, it would also be useful to have an asterisk or some other symbol to indicate significance to the readers quickly.

Experimental design

Despite the numerous positive attributes of the manuscript, aspects of the experimental design were of considerable concern, which ultimately led to my decision. The biggest source of my uncertainty is on the measurement of calcium presented here.

High technical standard of investigation:
In the study of crustacean exoskeleton composition, loss-by-ignition is not a common methodology. Common methods for elemental composition are energy-dispersive x-ray spectroscopy (EDX or EDS) for atomic or weight percent (deVries et al., 2016; Taylor et al., 2015; Wu and Zhou, 2011) as well as inductively-coupled plasma mass/atomic/optical emission spectrometry (ICP-MS or ICP-AS or ICP-OES) for exact concentrations (Eriander et al., 2015; Gravinese et al., 2016; Long et al., 2016; Page et al., 2016). Atomic adsorption spectrophotometry is also accepted (Findlay et al., 2010; France, 1987; Huner et al., 1978; Malley, 1980; Small et al., 2010; Wickins, 1984).

In this manuscript, it is unclear how the amount of calcium is determined from ash weight of the whole crayfish samples burned at one temperature. I am unable to access to the book used here as a reference, so I have looked to a small selection of primary literature for background on this method. It appears it is somewhat commonly used to analyze the organic carbon content and total carbon content sediment and soil. The sample is burned at around 500 deg C is designed to release the organic carbon, although some work has shown that there are multiple factors that can lead to some of that material remaining, necessitating a hotter exposure (Gallardo et al., 1987) or an exposure for at least 2.5 hours (Heiri et al., 2001). After organic matter is burned off and the sample weighed, the temperature is increased to around 900 deg C, which causes loss of the carbonates, and the sample must be weighed again to calculate this carbonate content (Dean Jr, 1974; Heiri et al., 2001). It is not described how carbonate (or, more specifically, how calcium from the carbonate amount) is determined in this research, but it appears that the data are presented with the assumption that the material left after an exposure of one hour at 500 deg C is 100% carbonate, which may not be true given that 1) the organic content is likely not fully burned off at this duration and temperature, and 2) the second burn exposure at 900 deg C and weighing was not carried out to specifically target carbonate compounds.

For biological samples where ash content or thermogravimetric analysis is used for one part of the analysis, additional methods are undertaken to determine elemental composition (Boßelmann et al., 2007; France, 1987; Monti et al., 2008; Rupérez, 2002), suggesting that burning off organic material is not enough to determine the amount of minerals present in the sample.

Furthermore, the data presented give me pause. The whole animal was combusted (or a homogenous sample if the animal was too large), and it seems unreasonable that, on average, this species of crayfish is comprised of ~50% mineral (Figure 4). In the crayfish species Orconecfes virilis, one study finds 12-17% of the carapace (where mineral is likely to be concentrated) is calcified (Malley, 1980), whereas later work finds 20-22% of the carapace and just 7-9% in the whole abdomen is comprised of calcium (France, 1987). In Procambarus clarkii, intermolt carapace calcium is 24% by weight (Huner et al., 1978).

To justify the methods presented as a way to accurately quantify the amount of calcium in a sample, please cite any examples where this methodology has been applied to biological samples, in particular whole crustaceans. Please write more details on this methodology, including how it is known that the result of one hour at 500 deg C results in carbonate alone, walk readers through the calculation of Ca from the measurement of carbonate, and describe how it is known that the mineral is CaCO3 rather than the sometimes-used MgCO3. Alternatively, if more work took place that negates these criticisms, provide more detail in the methods to explain these steps (eg digesting the ash in acid and performing atomic adsorption spectrophotometry).

This manuscript would also benefit by the addition of environmental data to describe the carbonate chemistry of the lakes to better support the relevance of the findings. The authors presented pH data at the site of collection. It is unclear if pH was sampled multiple times, but the pH would be better characterized with multiple measurements to capture the likely conditions under which these exoskeletons formed and mineralized. However, one other component of the carbonate system must also be measured (eg total alkalinity, pCO2, DIC), in addition to salinity and temperature, to accurately quantify the carbonate chemistry of the lake (Dickson et al., 2007). This will lend more support to the argument that lake conditions are the driving mechanism behind differences in crayfish calcium content. The crayfish biomineralization process may certainly be sensitive to the concentration of hydrogen ions in the external environment, but the saturation state of calcium carbonate minerals is also likely to play a role, although this cannot be determined from pH measurements alone.

Validity of the findings

Another concern of mine is that this research does not demonstrate a direct effect of pH on the differences in mineralization of the crayfish, although this is a focal topic of discussion. The difference in mineralization may be due to lake temperature, another undiscovered biological or abiotic factor, or simply different selection pressure for the crayfish in this lake. The relationship between lake pH and mineralization is purely speculation, but this might not come across to a casual reader, eg the title of this manuscript states that the pH level of this habitat has an effect on this species. Please be a little more restrained and limit the potential correlation between pH and mineralization to the discussion as one potential explanation for the results. Laboratory experiments that hold these crayfish in water of different pHs through at least one molt cycle would allow for a better explanation of these field findings.

Alternatively, consider refocusing the manuscript as a study on the differences in mineralization that one species can might achieve in the field; this would be a worthwhile addition to the crustacean exoskeleton field, as some researchers (most commonly engineers) who study the exoskeleton sample a single specimen as a representative of the whole species! Demonstrating spatial differences in exoskeleton composition would provide beneficial background for many researchers.

L226: There hasn’t been a demonstrated cost to a reduction in calcification. Less mineralization can reduce the density of the animal, for example, allowing it to achieve greater velocity through tail-flipping and avoid predators better in that regard. This study does not address the functional impacts of a change in mineralization, and so a hypothesis that there is a negative impact of reduced mineralization should be framed as such.

L230-231: For this research, it might be worthwhile to either discuss the possibility or reframe the results in the context that the reason Lake Ainsworth crayfish might have reduced calcium levels for one of two reasons: 1) as the authors state, this is the only lake that lands on the acidic side of the pH scale (simply chemistry, [H+] > [OH-], or 2) this pH level might simply be outside of the calcification tolerance of this species (biological control on the process of mineralization). In my research field, we see changes in calcification due to decreasing of ocean pH, although the oceans will never be acidic; instead, the seawater pH has moved outside of a range of tolerance.

Additional comments

Dear authors,

This research represents a timely focus on the adaptability of a nonnative crayfish species to multiple lakes with various abiotic conditions. The implications are well-described and provide a convincing argument for the study of characteristics that allow some organisms to adapt to a wide range of conditions.

Provided below are some more general comments as well as the literature references that I cited in my response.

Introduction
L66-70: Are these predators of this crayfish species? If not, I would recommend cutting theses lines. If yes, this might fit well with a discussion of how hard exoskeletons can help crustaceans deter biting predators (although there may be limited options for escape with such large predators!)

L69: In my opinion, this line seems a bit colored by the human experience rather than fact. Crayfish are a successful group of crustaceans and have evolved quite a few methods to avoid predation by large predators (eg the tail flip, the claws, coloration), and so they might not find these environments very challenging at all! I would think it is less likely to ruffle the feathers of any readers by changing this phrasing.

L81-82: Change the word “countries,” as the locations listed are not examples of countries.

Results
Line 194: It would be helpful to have either the range of calcium across locations reported here or the full means +/- sd for each location

Discussion
L202-203: This line does not properly highlight the novelty or contribution of this research. Please reframe to show these results.

L284-285: Here, I do not follow the logic of these species being a source of calcium. Would they not also have difficulty maintaining levels of calcium carbonate too if their mutual environment was a limiting factor?

References:

Bentov, S., Zaslansky, P., Al-Sawalmih, A., Masic, A., Fratzl, P., Sagi, A., Berman, A. and Aichmayer, B. (2012). Enamel-like apatite crown covering amorphous mineral in a crayfish mandible. Nature communications 3, 839.
Boßelmann, F., Romano, P., Fabritius, H., Raabek, D. and Epple, M. (2007). The composition of the exoskeleton of two crustacea: The American lobster Homarus americanus and the edible crab Cancer pagurus. Thermochimica Acta 463, 65--68.
Dean Jr, W. E. (1974). Determination of carbonate and organic matter in calcareous sediments and sedimentary rocks by loss on ignition: comparison with other methods. Journal of Sedimentary Research 44.
deVries, M. S., deVries, M. S., Webb, S. J., Tu, J., Cory, E., Morgan, V., Sah, R. L., Deheyn, D. D. and Taylor, J. R. A. (2016). Stress physiology and weapon integrity of intertidal mantis shrimp under future ocean conditions. Scientific Reports 6, 38637.
Dickson, A. G., Sabine, C. L. and Christian, J. R. (2007). Guide to best practices for ocean CO2 measurements. Sidney, BC: North Pacific Marine Science Organization.
Eriander, L., Wrange, A.-L. and Havenhand, J. (2015). Simulated diurnal pH fluctuations radically increase variance in—but not the mean of—growth in the barnacle Balanus improvisus. ICES Journal of Marine Science: Journal du Conseil, fsv214.
Findlay, H. S., Kendall, M. A., Spicer, J. I. and Widdicombe, S. (2010). Post-larval development of two intertidal barnacles at elevated CO2 and temperature. Marine Biology 157, 725-735.
France, R. L. (1987). Calcium and trace metal composition of crayfish (Orconectes virilis) in relation to experimental lake acidification. Canadian Journal of Fisheries and Aquatic Sciences 44, s107-s113.
Gallardo, J., Saavedra, J., Martin‐Patino, T. and Millan, A. (1987). Soil organic matter determination. Communications in soil science and plant analysis 18, 699-707.
Gravinese, P. M., Flannery, J. A. and Toth, L. T. (2016). A Methodology for Quantifying Trace Elements in the Exoskeletons of Florida Stone Crab (Menippe mercenaria) Larvae Using Inductively Coupled Plasma Optical Emission Spectrometry (ICP–OES): US Geological Survey.
Heiri, O., Lotter, A. F. and Lemcke, G. (2001). Loss on ignition as a method for estimating organic and carbonate content in sediments: reproducibility and comparability of results. Journal of paleolimnology 25, 101-110.
Huner, J. V., Kowalczuk, J. G. and Avault, J. W. (1978). Postmolt calcification in subadult red swamp crayfish, Procambarus clarkii (Girard)(Decapoda, Cambaridae). Crustaceana 34, 275-280.
Long, W. C., Swiney, K. M. and Foy, R. J. (2016). Effects of high pCO2 on Tanner crab reproduction and early life history, Part II: carryover effects on larvae from oogenesis and embryogenesis are stronger than direct effects. ICES Journal of Marine Science: Journal du Conseil 73, fsv251.
Malley, D. (1980). Decreased survival and calcium uptake by the crayfish Orconectes virilis in low pH. Canadian Journal of Fisheries and Aquatic Sciences 37, 364-372.
Monti, A., Di Virgilio, N. and Venturi, G. (2008). Mineral composition and ash content of six major energy crops. Biomass and Bioenergy 32, 216-223.
Page, T. M., Worthington, S., Calosi, P., Stillman, J. H. and Browman, H. e. H. (2016). Effects of elevated p CO2 on crab survival and exoskeleton composition depend on shell function and species distribution: a comparative analysis of carapace and claw mineralogy across four porcelain crab species from different habitats. ICES Journal of Marine Science 74, 1021-1032.
Raabe, D., Romano, P., Sachs, C., Fabritius, H., Al-Sawalmih, A., Yi, S. B., Servos, G. and Hartwig, H. G. (2006). Microstructure and crystallographic texture of the chitin–protein network in the biological composite material of the exoskeleton of the lobster Homarus americanus. Materials Science and Engineering: A 421, 143--153.
Raabe, D., Sachs, C. and Romano, P. (2005). The crustacean exoskeleton as an example of a structurally and mechanically graded biological nanocomposite material. Acta Materialia 53, 4281--4292.
Rupérez, P. (2002). Mineral content of edible marine seaweeds. Food chemistry 79, 23-26.
Small, D., Calosi, P., White, D., Spicer, J. I. and Widdicombe, S. (2010). Impact of medium-term exposure to CO2 enriched seawater on the physiological functions of the velvet swimming crab, Necora puber. Aquatic Biology 10, 11-21.
Taylor, J. R. A., Gilleard, J. M., Allen, M. C. and Deheyn, D. D. (2015). Effects of CO2-induced pH reduction on the exoskeleton structure and biophotonic properties of the shrimp Lysmata californica. Scientific Reports 5, 10608.
Wickins, J. (1984). The effect of hypercapnic sea water on growth and mineralization in penaied prawns. Aquaculture 41, 37 - 48.
Wu, Z. and Zhou, F. (2011). Structure and mechanical properties of pincers for freshwater lobster. Science China Technological Sciences 54, 650-658.

---

## Round 0.2 · Major Revisions

· Academic Editor

Major Revisions

Thank you for submitting your revision. All three reviewers were positive about the changes you have made. Based on their new feedback I am recommending major revisions to your current draft. I would like to invite you to submit a new revision along with a response to all three reviewers. While reviewer 2 comments that the importance of the study is limited, level of impact is not one of PeerJ’s editorial guidelines. However, other reviewers express concerns about the strength of experimental technical standards, support for conclusions and amount of conjecture in the discussion, which should be addressed.

I believe that these are some of the more important issues raised in the reviews that should be addressed:

1. Reviewer 3 expresses concerns about the water quality data not being collected at the same time as sample organisms and how the two historic samples from Lake Ainsworth are used.

2. Reviewer 3 has questions about the suitability of the loss by ignition technique and has suggestions for how it might be supported in the paper. Stronger support for this technique would be needed to satisfy PeerJ’s guidelines for technical standards.

3. All reviewers suggest revising the discussion to make it shorter to reduce the amount of speculation and better organized with stronger thesis statements in each paragraph.

4. I would consider removing the Lake Mooregah data as the sample size is low and variation so high (as suggested by reviewer 1). If you would like to make these data available to readers they could always be moved to a supplemental figure or table.

Please feel free to contact me if you have any questions as you make revisions and prepare a response to the reviewers.

Reviewer 1 ·

Basic reporting

This is second review, thus straight to constructive comments. The study reports valid results, but sampling has been very limited in 3 our 5 cases. The Discussions section is a bit confusing and should be compacted. The data from Lake Mooregah is not convinsing, and leaving it out would not take anything out of the rest of the data sets. Even the statistical comparisons would be valid.

Experimental design

See comments below.

Validity of the findings

Some notes on the results speculation:
1. Abstract and Results section claims Lake Ainsworth crayfish to differ significantly in their mineral content from other locations (Abstract, line 34) and from Emigrant Creeck Lake and Lake Somerset (Results, line 214). This is a bit confusing and should be presented in a similar manner in both cases.
2. Line 41 in Abstract, Lake mentioned but is there a word missing (Ainsworth)?
3. I would recommend using (or listing when first mentioned) common names, in addition to scientific names (line 93 onwards).
4. The authors are discussing possible reasons for observed differencies in mineral contents of the sampled populations. First, the sampled numbers are rather small in most cases (less than 5), which would suggest a rather limited speculation for the reasons. Also, the Discussions section is built with short, even one sentence paragraphs. I would suggest the authors to compact the presentation into a short note. As an example, the text in paragraphs starting from line 278 to line 303 compacted. The speculation here evolves around low pH and calcium, and could be presented in much compacted manner.
5. Lines 319-323, this sentence is very hard to understand and needs rewording. Seems like the authors are talking about mineralisation process and environmental (water quality) dynamics in this sentence, but in a manner that does not open to me.
6. Lines 326-331, in this paragraph there are two sentences, but it is hard to understand their logical connection. Word 'population' repeated here. Rewording needed.
7. Line 354, the pathway south is speculated to be some 600 km, what about the pathway north, some 600 km, too?
8. Lines 360-361, what are the authors trying to say here? It sounds strange, that the authors are doubting their data. I agree, though, that the sample numbers are too low, but are the authors stating it here?
9. Figure 2, the data from Lake Mooregah (n=2) does not look convinsing. I still suggest that this should be omitted, especially since the other too samples with low numbers do have much smaller SD (or variation).

Additional comments

The study is of value, but the presention should be compacted to a note. Most of the sample sizes are low (less than 5), which is normally considered preliminary data.

Reviewer 2 ·

Basic reporting

The revised manuscript is very much improved version as most of the comments have been accepted and incorporated into the revised version.
However, my only critical comment that would stay for this revised manuscript is that the research importance of this article is still limited and unfortunately nothing can be done to improve this unless there is some more data available that has not been used.

Experimental design

Greatly improved in terms of details now.

Validity of the findings

More references have been added

Additional comments

There is a great improvement in this version. The additional details in experimental design has made it more clearer now.
I still suggest to reduce some of the discussion points that are merely based on the speculation.

Reviewer 3 ·

Basic reporting

The revised introduction easily leads the reader through the background information on the species and provides compelling justification for the study. I do think the discussion could continue to benefit from some smoothing out, including reformation into paragraphs with clear topic sentences (in places such as 301 and 305, it appears single sentences are comprising a paragraph).

Experimental design

Thank you for being so thorough in explaining what you were measuring in your comments. I believe I better understand the differences in our lines of thinking now. From my understanding of your methods, after the organic carbon is burned off and the sample weighed again, the difference between dry weight and ash weight is considered to be total mineral content. I remain unconvinced that what remains is only mineral. I do not at all doubt that you have found differences in composition between these populations of crayfish, but I am not fully convinced that what remains after an hour exposure to 500 deg C is mineral content, especially mineral content that can be interpreted in the context of environmental influence on calcification.

I can imagine a few instances where this may not be the case. First, crustacean cuticle is a heterogenous material where mineral and organic material are tightly intermingled. As a paper I cited in my last review demonstrates (which was published after the protocol used here was established), sometimes an hour of ignition does not fully burn off organic carbon in sediment samples, and I am curious if that may be true for a crustacean cuticle, a material with a complex biological matrix of organic content and mineral. I have not myself examined ashes for evidence of incomplete combustion and do not know if visual inspection may identify if there is any remaining organic material on the micro scale of the matrix.

Secondly, it is my understanding that other inorganic elements and compounds will remain after this ignition protocol, which are included in the weight measurement going into calculations of the mineral content but are not strictly minerals. These may include metals that are found in crustacean cuticles, such as aluminum and manganese, and other elements such as chlorine. If one sampling location has more bioavailable metals (lower pH waters may increase bioavailability), the differences in the weights of your samples may have arisen for this reason, not because of the available calcifying elements or pH. In short, I have not yet found any other papers that use only the data from a loss of organic carbon to discuss mineralogy.

I will be more satisfied that these methods are sufficient for producing the desired result of mineral content if one or all of the following are provided: 1) peer-reviewed literature that demonstrates that these methods have been used on biological samples to quantify mineralogy, particularly for samples that include both organic content and mineral in a heterogenous material (especially crustaceans). 2) If the ashes are still accessible, determining carbonate content for the reasons I suggested in my last review (provides more specificity for quantifying the calcium carbonate polymorphs, although you would not be able to calculate any calcium associated with hydroxyapatite) or using another technique to measure the ashes for calcium and magnesium to compare to the water sample data. I do not believe knowing the exact concentrations of the minerals in the samples are required, but this approach would assuage any doubts that your methods are only targeting the calcifying minerals. 3) Using new samples to demonstrate that another established technique of measuring mineral content and this protocol produce analogous results. 4) I’d also be amenable to another option. In short, I would like to be assured that the waterbody-specific differences in percent weight of the remaining material is indeed mineral likely to have been used in calcification (eg calcium carbonate) and not another non-organic component of the crayfish bodies. This will better support the argument that the available waterbody data does provide insight into the differences in the composition of these crayfish.

I also do have to mention a potential sticking point with the water data. I appreciate the clear effort put into procuring and organizing it for this revision. I am concerned that these samples from Lake Ainsworth were taken 18+ years before the crayfish samples were collected, and as line 227 notes, crayfish were not present in this waterbody until more than a decade after. A lot may have changed in the water conditions in intervening years that is not captured here. Ideally, more recent calcium and magnesium measurements could be made, although it has now been a few years since samples were collected, but this would potentially provide evidence that conditions have been stable. However, if the 1996 and 1987 samples were taken from the same area of the lake, or if the lake is small enough to be homogenous , there seems to be evidence that there are large changes in lake minerals. It is promising that there are not large differences in the pH you have measured and the pH from these samples. Also, thank you for being upfront about this in the figure caption. If you are not able to get more recent data, I suggest also noting the difference in the age of samples in the text as a caveat also.

What does seem more problematic is that parts of Lake Ainsworth have (or used to have) a very high amount of calcium—the maximum values are 3-4 times greater than the maximums recorded for other waterbodies. Please justify the focus on the 1996 samples rather than the 1987 samples which include high measurements of calcium that do not support the hypothesis presented here.

Validity of the findings

Mixed in with above

Additional comments

Thank you for your revisions and well-organized comments. Please find a few remaining line comments below.

Methods
Line 142: To clarify my last comment on the pH scale, I was asking whether pH was measured on the total, NBS, or free scale. These scales can be more than 0.1 pH units different from each other. If you are only interested in the differences in pH across sites, this might not be important, but I think it would be useful to include to add context to the absolute values in Table 2 and knowing if the data in Table 1 and Table 2 are measured on the same scale is important when comparing them. I believe NBS is most common for freshwater.

Results
Line 210: To clarify my last comment on line 194 of the first manuscript, I would appreciate seeing the means +/- SE of the crayfish mineral content here in the results (potentially in Table 1) rather than trying to read values off of Figure 2. I’m sorry for being unclear about that.

Discussion
Line 288: It might not be clear to readers how acidity and low calcium levels may be two separate issues based on the information provided in above paragraphs that focus on how calcareous mineral or calcium storage and deposition are affected. I believe it would be clearer to also mention that bicarbonate is less available at reduced pHs, thereby making calcification more difficult if bicarbonate isn’t fully supplied physiologically. Thus, if acidity and low calcium levels interact, both components of calcium carbonate may be difficult to procure.

Line 309: I believe this introduction of cuticle thickness needs more context or should be left out. Yes, these papers also studied crayfish living in low pH environments, but since the response variable studied is different, I do not think they are very complementary. Cuticle thickness may not be correlated with mineral content, as mineral density might differ, and there may be different mechanisms behind more dense or reduced construction of Bouligand layers than impregnation of mineral. Without further explanation, readers may be thrown off by a somewhat abrupt injection of discussion of structure rather than composition.

---

## Round 0.3 · Minor Revisions

· Academic Editor

Minor Revisions

Thank you for submitting your revised work to PeerJ. I am recommending minor revisions at this point to address what I think are still some outstanding issues. I do not plan to send any resubmission back to reviewers as I believe that I will be able to assess how these issues have been addressed.

I first want to thank you for your thorough revisions and work to address reviewer comments. There are three issues that I would like to ask you to address before your manuscript could be accepted for publication:

1. Your manuscript now provides a good description of what is being measured with the loss-by-ignition method. Your revised Methods section on page 7 of the marked-up text now explains that this technique measures incombustible residues, which may include materials in addition to minerals. This is a valuable addition to the Methods, and is reiterated in the new Limitations section towards the end of the Discussion. However, the Methods section then states that this ash is “considered as a measure of the mineral content of the samples” and the term “mineral content” is used throughout the paper, including in the abstract. But as you describe it, this mass is not just mineral content. Your Limitations section states that the technique resulted in a “consistent measure of actual minerals, plus all non-targets, in all samples”. But it seems possible that total ash might not scale with mineral content if non-mineral components differ between the sample locations. As written your use of “mineral content” may be misleading for readers, even with your explanation in the Methods. I hope you agree that it is more accurate to refer to this remaining mass as “incombustible material” throughout the manuscript.

I understand your explanation of the validity of the loss-by-ignition method, but it would still be helpful to cite examples of its use in analyzing crustacean mineral content. If such studies do not exist I think it is even more important to refer to remaining mass as “incombustible material”.

2. Considering the points in comment 1 above I recommend rephrasing the discussion to refer to what has been measured as “incombustible material” and make it clear that an assumption is being made that this measurement reflects mineral content. I believe that making this caveat clear is important for readers before they consider your discussion of how water conditions in Lake Ainsworth may have affected total ash content in specimens from that location. I believe that it would also be important to frame the discussion about the effect of Lake Ainsworth water conditions on crayfish mineralization as a hypothesis arising from your data that requires further data collection. Those future data would include current water mineral content in the Lake and a specific analysis of crayfish calcium content.

3. The addition of the note to the legend for Table 2 helps clarify how the historical data for Lake Ainsworth are being used. The two historical data points for this lake do seem to provide support for the consistently low pH at this site. These historical data are also explained in a new final sentence in the Limitations section. I’d like to ask that this sentence provide more specifics explaining how these data are helpful (I believe by showing long term low pH) and also explain why you believe some of the data are outliers and not included in your consideration of the Lake’s effects on its crayfish population.

Please feel free to contact me if you have questions while addressing these points. I look forward to receiving your resubmission.

Reviewer 3 ·

Basic reporting

Addressed with previous reviews

Experimental design

Comments below

Validity of the findings

Comments below

Additional comments

Dear authors,

Thank you for the revision and working to clarify some points. Here are my short thoughts:

With this revision, we agree on the results provided from this methodology, and I am glad we have reached this understanding. Yet, this now means I don’t follow the conclusions from these data. Because we both agree that the method applied does not provide quantification of the constituents of the material left over from combustion, I do not understand how the discussion may speculate extensively about how the conditions of the lakes influence a specific part of the mineral content measured. Authors, I do understand that your goal was to measure “the amount (quantified by DWT) of incombustible material in the crayfish.” Yet, the discussion does not stop here—I would be satisfied if it did, simply pointing out these differences. Just your crayfish data and the assembled water quality, separately, do add valuable information to the literature, as you assert in your last response. However, potential calcium differences in these crayfish feature heavily in the discussion—about five paragraphs’ worth—although was not studied by these methodologies, as we agree.

I’m not sure how else to bridge this gap, so please allow me to present how I see this situation, in another context. Researchers set out to study five forests and record the total abundance of furry animals (total mineral content) in each, and they find some forests have fewer than others. Discussing the results, they argue that the number of squirrels in some forests are reduced, probably because there are fewer oak trees. This is the breakdown I see; if the researchers wanted to discuss squirrels, that is the data that should have been collected (this was my sticking point on the last two reviews—why calcium content wasn’t measured when that was the component of the incombustible material that was of interest). Perhaps there are the same number of squirrels in the forests, and the differences in the number of furry animals is instead due to the number of rats (metals, for example); there’s no way to know that from just the data on furry animals alone. Or, more troubling, the differences in furry animals are due to a few animals; there are fewer squirrels in one forest and fewer rats in another—this is something else the methodology cannot parse apart. Changes in total mineral content do not provide information on how calcium differs in the body of these crayfish, as other constituents may be driving the differences in Lake Ainsworth crayfish.
I do see great value in publishing the basic data (in a reduced manuscript with less speculative discussion, as reviewers 1 and 2 have brought up), as you argued in the latest response, and I appreciate the work you as authors have put in; the changes I have seen over these few revisions have led to a stronger, refined piece of writing. However, I remain unconvinced about the how differences in total mineral content arise.

---

## Round 0.4 · accepted · Accept

· Academic Editor

Accept

Thank you for addressing my last set of comments in your current revision. I am happy to now accept your manuscript for publication in PeerJ. I have two minor suggestions for changes that you could make when submitting the draft during production.

There is an errant open parenthesis in line 362

“Calcium” in lines 419-421 should be lower case

You will be given the option to make the reviews of your manuscript available to readers. Please consider doing so as this review record can be a great resource for readers of your paper and contributes to more transparent science.

Thank you again for your contribution.

#